# NAGPHORMER: A TOKENIZED GRAPH TRANSFORMER FOR NODE CLASSIFICATION IN LARGE GRAPHS

**Jinsong Chen**[1,2,3,*]**, Kaiyuan Gao**[2,3,*]**, Gaichao Li**[1,2,3]**, Kun He**[2,3,†]
[1]Institute of Artificial Intelligence, Huazhong University of Science and Technology
[2]School of Computer Science and Technology, Huazhong University of Science and Technology
[3]Hopcroft Center on Computing Science, Huazhong University of Science and Technology
  {chenjinsong,im_kai,gaichaolee,brooklet60}@hust.edu.cn

## ABSTRACT

The graph Transformer emerges as a new architecture and has shown superior performance on various graph mining tasks. In this work, we observe that existing graph Transformers treat nodes as independent tokens and construct a single long sequence composed of all node tokens so as to train the Transformer model, causing it hard to scale to large graphs due to the quadratic complexity on the number of nodes for the self-attention computation. To this end, we propose a Neighborhood Aggregation Graph Transformer (NAGphormer) that treats each node as a sequence containing a series of tokens constructed by our proposed Hop2Token module. For each node, Hop2Token aggregates the neighborhood features from different hops into different representations and thereby produces a sequence of token vectors as one input. In this way, NAGphormer could be trained in a mini-batch manner and thus could scale to large graphs. Moreover, we mathematically show that as compared to a category of advanced Graph Neural Networks (GNNs), the decoupled Graph Convolutional Network, NAGphormer could learn more informative node representations from the multi-hop neighborhoods. Extensive experiments on benchmark datasets from small to large are conducted to demonstrate that NAGphormer consistently outperforms existing graph Transformers and mainstream GNNs. Code is available at https://github.com/JHL-HUST/NAGphormer.

## 1 INTRODUCTION

Graphs, as a powerful data structure, are widely used to represent entities and their relations in a variety of domains, such as social networks in sociology and protein-protein interaction networks in biology. Their complex features (*e.g.*, attribute features and topology features) make the graph mining tasks very challenging. Graph Neural Networks (GNNs) (Chen et al., 2020c; Kipf & Welling, 2017; Veličković et al., 2018), owing to the message passing mechanism that aggregates neighborhood information for learning the node representations (Gilmer et al., 2017), have been recognized as a type of powerful deep learning techniques for graph mining tasks (Xu et al., 2019; Fan et al., 2019; Ying et al., 2018; Zhang & Chen, 2018; Jin et al., 2019) over the last decade. Though effective, message passing-based GNNs have a number of inherent limitations, including over-smoothing (Chen et al., 2020a) and over-squashing (Alon & Yahav, 2021) with the increment of model depth, limiting their potential capability for graph representation learning. Though recent efforts (Yang et al., 2020; Lu et al., 2021; Huang et al., 2020; Sun et al., 2022) have been devoted to alleviate the impact of over-smoothing and over-squashing problems, the negative influence of these inherent limitations cannot be eliminated completely.

Transformers (Vaswani et al., 2017), on the other hand recently, are well-known deep learning architectures that have shown superior performance in a variety of data with an underlying Euclidean or grid-like structure, such as natural languages (Devlin et al., 2019; Liu et al., 2019) and images (Dosovitskiy et al., 2021; Liu et al., 2021). Due to their great modeling capability, there is a growing interest in generalizing Transformers to non-Euclidean data like graphs (Dwivedi & Bresson, 2020;

---

[*]The first two authors contribute equally.
[†]Corresponding author.

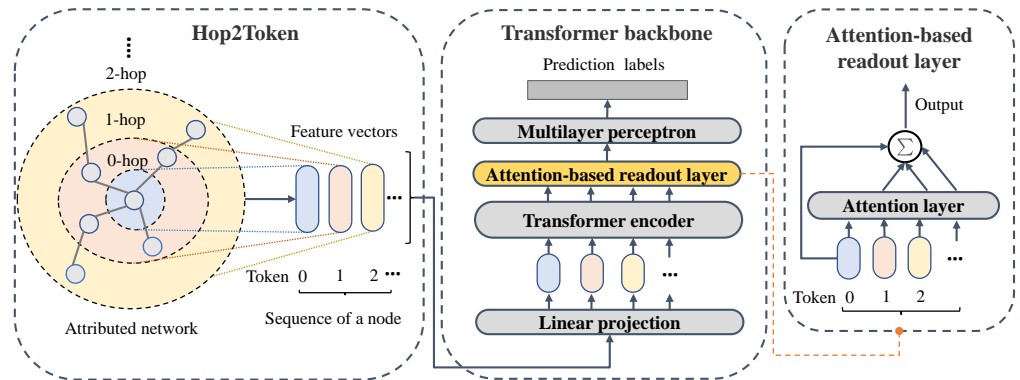

Figure 1: **Model framework of NAGphormer.** NAGphormer first uses a novel neighborhood aggregation module, Hop2Token, to construct a sequence for each node based on the tokens of different hops of neighbors. Then, NAGphormer learns the node representations using a Transformer backbone, and an attention-based readout function is developed to aggregate neighborhood information of different hops adaptively. An MLP-based module is used in the end for label prediction.

Kreuzer et al., 2021; Ying et al., 2021; Jain et al., 2021). However, graph-structured data generally contain more complicated properties, including structural topology and attribute features, that cannot be directly encoded into Transformers as the tokens.

Existing graph Transformers have developed three techniques to address this issue (Min et al., 2022): introducing structural encoding (Dwivedi & Bresson, 2020; Kreuzer et al., 2021), using GNNs as auxiliary modules (Jain et al., 2021), and incorporating graph bias into the attention matrix (Ying et al., 2021). By integrating structural information into the model, graph Transformers exhibit competitive performance on various graph mining tasks, and outperform GNNs on node classification (Kreuzer et al., 2021; Chen et al., 2022) and graph classification (Ying et al., 2021; Jain et al., 2021) tasks on small to mediate scale graphs.

In this work, we observe that existing graph Transformers treat the nodes as independent tokens and construct a single sequence composed of all the node tokens to train the Transformer model, causing a quadratic complexity on the number of nodes for the self-attention calculation. Training such a model on large graphs will cost a huge amount of GPU resources that are generally unaffordable since the mini-batch training is unsuitable for graph Transformers using a single long sequence as the input. Meanwhile, effective strategies that make GNNs scalable to large-scale graphs, including node sampling (Chen et al., 2018; Zou et al., 2019) and approximation propagation (Chen et al., 2020b; Feng et al., 2022), are not applicable to graph Transformers, as they capture the global attention of all node pairs and are independent of the message passing mechanism. The current paradigm of graph Transformers makes it intractable to generalize to large graphs.

To address the above challenge, we propose a novel model dubbed Neighborhood Aggregation Graph Transformer (NAGphormer) for node classification in large graphs. Unlike existing graph Transformers that regard the nodes as independent tokens, NAGphormer treats each node as a sequence and constructs tokens for each node by a novel neighborhood aggregation module called Hop2Token. The key idea behind Hop2Token is to aggregate neighborhood features from multiple hops and transform each hop into a representation, which could be regarded as a token. Hop2Token then constructs a sequence for each node based on the tokens in different hops to preserve the neighborhood information. The sequences are then fed into a Transformer-based module for learning the node representations. By treating each node as a sequence of tokens, NAGphormer could be trained in a mini-batch manner and hence can handle large graphs even on limited GPU resources.

Considering that the contributions of neighbors in different hops differ to the final node representation, NAGphormer further provides an attention-based readout function to learn the importance of each hop adaptively. Moreover, we provide theoretical analysis on the relationship between NAGphormer and an advanced category of GNNs, the decoupled Graph Convolutional Network (GCN) (Dong et al., 2021; Klicpera et al., 2019; Wu et al., 2019; Chien et al., 2021). The analysis is from the

perspective of self-attention mechanism and Hop2Token, indicating that NAGphormer is capable of learning more informative node representations from the multi-hop neighborhoods. We conduct extensive experiments on various popular benchmarks, including six small datasets and three large datasets, and the results demonstrate the superiority of the proposed method.

The main contributions of this work are as follows:

- We propose Hop2Token, a novel neighborhood aggregation method that aggregates the neighborhood features from each hop into a node representation, resulting in a sequence of token vectors that preserves neighborhood information for different hops. In this way, we can regard each node in the complex graph data as a sequence of tokens, and treat them analogously as in natural language processing and computer vision fields.

- We propose a new graph Transformer model, NAGphormer, for the node classification task. NAGphormer can be trained in a mini-batch manner depending on the output of Hop2Token, and therefore enables the model to handle large graphs. We also develop an attention-based readout function to adaptively learn the importance of different-hop neighborhoods to further boost the model performance.

- We prove that from the perspective of self-attention mechanism, the proposed NAGphormer can learn more expressive node representations from the multi-hop neighborhoods compared to an advanced category of GNNs, the decoupled GCN.

- Extensive experiments on benchmark datasets from small to large demonstrate that NAGphormer consistently outperforms existing graph Transformers and mainstream GNNs.

## 2 BACKGROUND

### 2.1 PROBLEM FORMULATION

Let $G = (V, E)$ be an unweighted and undirected attributed graph, where $V = \{v_1, v_2, \cdots, v_n\}$, and $n = |V|$. Each node $v \in V$ has a feature vector $\mathbf{x}_v \in \mathbf{X}$, where $\mathbf{X} \in \mathbb{R}^{n \times d}$ is the feature matrix describing the attribute information of nodes and $d$ the dimension of feature vector. $\mathbf{A} \in \mathbb{R}^{n \times n}$ represents the adjacency matrix and $\mathbf{D}$ the diagonal degree matrix. The normalized adjacency matrix is defined as $\hat{\mathbf{A}} = \tilde{\mathbf{D}}^{-1/2} \tilde{\mathbf{A}} \tilde{\mathbf{D}}^{-1/2}$, where $\tilde{\mathbf{A}}$ denotes the adjacency matrix with self-loops and $\tilde{\mathbf{D}}$ denotes the corresponding degree matrix. The node classification task provides a labeled node set $V_l$ and an unlabeled node set $V_u$. Let $\mathbf{Y} \in \mathbb{R}^{n \times c}$ denote the label matrix where $c$ is the number of classes. Given the labels $\mathbf{Y}_{V_l}$, the goal is to predict the labels $\mathbf{Y}_{V_u}$ for unlabeled nodes.

### 2.2 GRAPH NEURAL NETWORK

Graph Neural Network (GNN) has become a powerful technique to model the graph-structured data. Graph Convolutional Network (GCN) (Kipf & Welling, 2017) is a typical model of GNN that applies the first-order approximation of spectral convolution (Defferrard et al., 2016) to aggregate information of immediate neighbors. A GCN layer can be written as:

$$\mathbf{H}^{(l+1)} = \sigma(\hat{\mathbf{A}} \mathbf{H}^{(l)} \mathbf{W}^{(l)}), \tag{1}$$

where $\mathbf{H}^{(l)} \in \mathbb{R}^{n \times d^{(l)}}$ and $\mathbf{W}^{(l)} \in \mathbb{R}^{d^{(l)} \times d^{(l+1)}}$ denote the representation of nodes and the learnable parameter matrix in the $l$-th layer, respectively. $\sigma(\cdot)$ denotes the non-linear activation function.

Equation 1 contains two operations, neighborhood aggregation and feature transformation, which are coupled in the GCN layer. Such a coupled design would lead to the over-smoothing problem (Chen et al., 2020a) when the number of layers increases, limiting the model to capture deep structural information. To address this issue, the decoupled GCN (Klicpera et al., 2019; Wu et al., 2019) separates the feature transformation and neighborhood aggregation in the GCN layer and treats them as independent modules. A general form of decoupled GCN is described as (Chien et al., 2021):

$$\mathbf{Z} = \sum_{k=0}^{K} \beta_k \mathbf{H}^{(k)}, \mathbf{H}^{(k)} = \hat{\mathbf{A}} \mathbf{H}^{(k-1)}, \mathbf{H}^{(0)} = \boldsymbol{f}_\theta(\mathbf{X}), \tag{2}$$

where $\mathbf{Z}$ denotes the final representations of nodes, $\mathbf{H}^{(k)}$ denotes the hidden representations of nodes at propagation step $k$, $\beta_k$ denotes the aggregation coefficient of propagation step $k$, $\hat{\mathbf{A}}$ denotes the normalized adjacency matrix, $\boldsymbol{f}_\theta$ denotes a neural network module and $\mathbf{X}$ denotes the raw attribute feature matrix. Such a decoupled design exhibits high computational efficiency and enables the model capture deeper structural information. More related works on GNNs are provided in Appendix A.

## 2.3 TRANSFORMER

The Transformer encoder (Vaswani et al., 2017) contains a sequence of Transformer layers, where each layer is comprised with a multi-head self-attention (MSA) and a position-wise feed-forward network (FFN). The MSA module is the critical component that aims to capture the semantic correlation between the input tokens. For simplicity, we use the single-head self-attention module for description. Suppose we have an input $\mathbf{H} \in \mathbb{R}^{n \times d}$ for the self-attention module where $n$ is the number of tokens and $d$ the hidden dimension. The self-attention module first projects $\mathbf{H}$ into three subspaces, namely $\mathbf{Q}$, $\mathbf{K}$ and $\mathbf{V}$:

$$\mathbf{Q} = \mathbf{H}\mathbf{W}^Q, \ \mathbf{K} = \mathbf{H}\mathbf{W}^K, \ \mathbf{V} = \mathbf{H}\mathbf{W}^V, \tag{3}$$

where $\mathbf{W}^Q \in \mathbb{R}^{d \times d_K}, \mathbf{W}^K \in \mathbb{R}^{d \times d_K}$ and $\mathbf{W}^V \in \mathbb{R}^{d \times d_V}$ are the projection matrices. The output matrix is calculated as:

$$\mathbf{H}' = \mathrm{softmax}\left(\frac{\mathbf{Q}\mathbf{K}^\top}{\sqrt{d_K}}\right)\mathbf{V}. \tag{4}$$

The attention matrix, $\mathrm{softmax}\left(\frac{\mathbf{Q}\mathbf{K}^\top}{\sqrt{d_K}}\right)$, captures the pair-wise similarity of input tokens in the sequence. Specifically, it calculates the dot product between each token pair after projection. The softmax is applied row-wise.

**Graph Transformer.** The Transformer architecture has attracted increasing attention in graph representation learning in recent years. The key idea of graph Transformer is to integrate graph structural information to the Transformer architecture so as to learn the node representations. Existing graph Transformers could be divided into three categories: (I) Replace the positional encoding with Laplacian eigenvectors (Dwivedi & Bresson, 2020; Kreuzer et al., 2021) or degree-related feature vectors (Ying et al., 2021) to capture the structural features of nodes. (II) In addition to positional encoding, GNNs are used as auxiliary modules to enable the Transformer model to capture structural information (Jain et al., 2021; Rong et al., 2020). (III) Introduce graph information bias into the attention score of each node pair, *e.g.*, the shortest-path distance (Ying et al., 2021). We provide a detailed review of graph Transformer in Appendix A.

## 3 THE PROPOSED NAGPHORMER

In this section, we present the proposed NAGphormer in details. To handle graphs at scale, we first introduce a novel neighborhood aggregation module called Hop2Token, then we build NAGphormer together with structural encoding and attention-based readout function.

### 3.1 HOP2TOKEN

How to aggregate information from adjacent nodes into a node representation is crucial in reasonably powerful Graph Neural Network (GNN) architectures. To inherit the desirable properties, we design Hop2Token considering the neighborhood information of different hops.

For a node $v$, let $\mathcal{N}^k(v) = \{u \in V | d(v, u) \leq k\}$ be its $k$-hop neighborhood, where $d(v, u)$ represents the distance of shortest path between $v$ and $u$. We define $\mathcal{N}^0(v) = \{v\}$, *i.e.*, the 0-hop neighborhood is the node itself. In Hop2Token, we transform the $k$-hop neighborhood $\mathcal{N}^k(v)$ into a neighborhood embedding $\mathbf{x}_v^k$ with an aggregation operator $\phi$. In this way, the $k$-hop representation of a node $v$ can be expressed as:

$$\mathbf{x}_v^k = \phi(\mathcal{N}^k(v)). \tag{5}$$

By Equation 5, we can calculate the neighborhood embeddings for variable hops of a node and further construct a sequence to represent its neighborhood information, *i.e.*, $\mathcal{S}_v = (\mathbf{x}_v^0, \mathbf{x}_v^1, ..., \mathbf{x}_v^K)$, where $K$ is fixed as a hyperparameter. Assume $\mathbf{x}_v^k$ is a $d$-dimensional vector, the sequences of all nodes

---

**Algorithm 1** The Hop2Token Algorithm

---

**Input:** Normalized adjacency matrix $\hat{\mathbf{A}}$; Feature matrix $\mathbf{X}$; Propagation step $K$
**Output:** Sequences of all nodes $\mathbf{X}_G$
  1: **for** $k = 0$ to $K$ **do**
  2:     **for** $i = 0$ to $n$ **do**
  3:         $\mathbf{X}_G[i, k] = \mathbf{X}[i]$;
  4:     **end for**
  5:     $\mathbf{X} = \hat{\mathbf{A}}\mathbf{X}$;
  6: **end for**
  7: **return** Sequences of all nodes $\mathbf{X}_G$;

---

in graph $G$ will construct a tensor $\mathbf{X}_G \in \mathbb{R}^{n \times (K+1) \times d}$. To better illustrate the implementation of Hop2Token, we decompose $\mathbf{X}_G$ to a sequence $\mathcal{S} = (\mathbf{X}_0, \mathbf{X}_1, \cdots, \mathbf{X}_K)$, where $\mathbf{X}_k \in \mathbb{R}^{n \times d}$ can be seen as the $k$-hop neighborhood matrix. Here we define $\mathbf{X}_0$ as the original feature matrix $\mathbf{X}$.

In practice, we apply a propagation process similar to the method in (Chien et al., 2021; He et al., 2022) to obtain the sequence of $K$-hop neighborhood matrices. Given the normalized adjacency matrix $\hat{\mathbf{A}}$ (*aka* the transition matrix (Gasteiger et al., 2019)) and $\mathbf{X}$, multiplying $\hat{\mathbf{A}}$ with $\mathbf{X}$ aggregates immediate neighborhood information. Applying this multiplication consecutively allows us to propagate information at larger distances. For example, we can access 2-hop neighborhood information by $\hat{\mathbf{A}}(\hat{\mathbf{A}}\mathbf{X})$. Thereafter, the $k$-hop neighborhood matrix can be described as:

$$\mathbf{X}_k = \hat{\mathbf{A}}^k \mathbf{X}. \tag{6}$$

The detailed implementation is drawn in Algorithm 1. The advantages of Hop2Token is two-fold. (I) Hop2Token is a non-parametric method. It can be conducted offline before the model training, and the output of Hop2Token supports mini-batch training. In this way, the model can handle graphs of arbitrary sizes, thus allowing the generalization of graph Transformer to large-scale graphs. (II) Encoding $k$-hop neighborhood of a node into one representation is helpful for capturing the hop-wise semantic correlation, which is ignored in typical GNNs (Kipf & Welling, 2017; Klicpera et al., 2019; Chien et al., 2021).

### 3.2 NAGphormer for Node Classification

Figure 1 depicts the architecture of NAGphormer. Given an attributed graph, we first concatenate a matrix constructed by eigendecomposition to the attribute matrix, and gain a structure-aware feature matrix. Accordingly, the effective feature vector for node $v$ is extended as $\mathbf{x}_v \in \mathbb{R}^{d'}$. The detailed construction is described in Section 3.3.

Next, we assemble an aggregated neighborhood sequence as $\mathcal{S}_v = (\mathbf{x}_v^0, \mathbf{x}_v^1, ..., \mathbf{x}_v^K)$ by applying Hop2Token. Then we map $\mathcal{S}_v$ to the hidden dimension $d_m$ of the Transformer with a learnable linear projection:

$$\mathbf{Z}_v^{(0)} = \left[ \mathbf{x}_v^0 \mathbf{E}; \ \mathbf{x}_v^1 \mathbf{E}; \ \cdots ; \ \mathbf{x}_v^K \mathbf{E} \right], \tag{7}$$

where $\mathbf{E} \in \mathbb{R}^{d' \times d_m}$ and $\mathbf{Z}_v^{(0)} \in \mathbb{R}^{(K+1) \times d_m}$.

Then, we feed the projected sequence into the Transformer encoder. The building blocks of the Transformer contain multi-head self-attention (MSA) and position-wise feed-forward network (FFN). We follow the implementation of the vanilla Transformer encoder described in (Vaswani et al., 2017), while LayerNorm (LN) is applied before each block (Xiong et al., 2020). And the FFN consists of two linear layers with a GELU non-linearity:

$$\mathbf{Z}_v'^{(\ell)} = \text{MSA}\left( \text{LN}\left( \mathbf{Z}_v^{(\ell-1)} \right) \right) + \mathbf{Z}_v^{(\ell-1)}, \tag{8}$$

$$\mathbf{Z}_v^{(\ell)} = \text{FFN}\left( \text{LN}\left( \mathbf{Z}_v'^{(\ell)} \right) \right) + \mathbf{Z}_v'^{(\ell)}, \tag{9}$$

where $\ell = 1, \ldots, L$ implies the $\ell$-th layer of the Transformer.

In the end, a novel readout function is applied to the output of the Transformer encoder. Through several Transformer layers, the corresponding output $\mathbf{Z}_v^{(\ell)}$ contains the embeddings for all neighborhoods

of node $v$. It requires a readout function to aggregate the information of different neighborhoods into one embedding. Common readout functions include summation and mean (Hamilton et al., 2017). However, these methods ignore the importance of different neighborhoods. Inspired by GAT (Veličković et al., 2018), we propose an attention-based readout function to learn such importance by computing the attention coefficients between 0-hop neighborhood (*i.e.*, the node itself) and every other neighborhood. For detailed implementation, please refer to Section 3.3.

The time and space complexity of NAGphormer are $O(n(K + 1)^2 d)$ and $O(b(K + 1)^2 + b(K + 1)d + d^2 L)$, respectively ($n$: number of nodes, $K$: number of hops, $d$: dimension of feature vector, $L$: number of layers, $b$: batch size). The detailed complexity analysis is provided in Appendix B.

### 3.3 IMPLEMENTATION DETAILS

**Structural encoding.** Besides the attribute information of nodes, the structural information of nodes is also a crucial feature for graph mining tasks. We adopt the eigenvectors of Laplacian matrix of the graph for capturing the structural information of nodes. Specifically, we select the eigenvectors corresponding to the $s$ smallest non-trivial eigenvalues to construct the structure matrix $\mathbf{U} \in \mathbb{R}^{n \times s}$ (Dwivedi & Bresson, 2020; Kreuzer et al., 2021). Then we combine the original feature matrix $\mathbf{X}$ with the structure matrix $\mathbf{U}$ to preserve both the attribute and structural information:

$$\mathbf{X}' = \mathbf{X} \| \mathbf{U}. \tag{10}$$

Here $\|$ indicates the concatenation operator and $\mathbf{X}' \in \mathbb{R}^{n \times (d+s)}$ denotes the fused feature matrix, which is then used as the input of Hop2Token for calculating the information of different-hop neighborhoods.

**Attention-based readout function.** For the output matrix $\mathbf{Z} \in \mathbb{R}^{(K+1) \times d_m}$ of a node, $\mathbf{Z}_0$ is the token representation of the node itself and $\mathbf{Z}_k$ is its $k$-hop representation. We calculate the normalized attention coefficients for its $k$-hop neighborhood:

$$\alpha_k = \frac{exp((\mathbf{Z}_0 \| \mathbf{Z}_k) \mathbf{W}_a^\top)}{\sum_{i=1}^{K} exp((\mathbf{Z}_0 \| \mathbf{Z}_i) \mathbf{W}_a^\top)}, \tag{11}$$

where $\mathbf{W}_a \in \mathbb{R}^{1 \times 2d_m}$ denotes the learnable projection and $i = 1, \ldots, K$. Therefore, the readout function takes the correlation between each neighborhood and the node representation into account. The node representation is finally aggregated as follows:

$$\mathbf{Z}_{out} = \mathbf{Z}_0 + \sum_{k=1}^{K} \alpha_k \mathbf{Z}_k. \tag{12}$$

### 3.4 THEORETICAL ANALYSIS OF NAGPHORMER

In this subsection, we discuss the relation of NAGphormer and decoupled GCN through the lens of the node representations of Hop2Token and the self-attention mechanism. We theoretically show that NAGphormer could learn more informative node representations from the multi-hop neighborhoods than decoupled GCN does.

**Fact 1.** *From the perspective of the output node representations of Hop2Token, we can regard the decoupled GCN as applying a self-attention mechanism with a fixed attention matrix $\mathbf{S} \in \mathbb{R}^{(K+1) \times (K+1)}$, where $\mathbf{S}_{K,k} = \beta_k$ ($k \in \{0, ..., K\}$) and other elements are all zeroes.*

Here $K$ denotes the total propagation step, $k$ represents the current propagation step, $\beta_k$ represents the aggregation weight at propagation step $k$ in the decoupled GCN. The detailed proof of **Fact 1** is provided in Appendix C. **Fact 1** indicates that the decoupled GCN, an advanced category of GNN, only captures partial information of the multi-hop neighborhoods through the incomplete attention matrix. Moreover, the fixed attention coefficients of $\beta_k$ ($k \in \{0, ..., K\}$) for all nodes also limit the model to learn the node representations adaptively from their individual neighborhood information.

In contrast, our proposed NAGphormer first utilizes the self-attention mechanism to learn the representations of different-hop neighborhoods based on their semantic correlation. Then NAGphormer develops an attention-based readout function to adaptively learn the node representations from their neighborhood information, which helps the model learn more informative node representations.

Table 1: Comparison of all models in terms of mean accuracy $\pm$ stdev (%) on small-scale datasets. The best results appear in **bold**. OOM indicates the out-of-memory error.

| Method | Pubmed | CoraFull | Computer | Photo | CS | Physics |
|---|---|---|---|---|---|---|
| GCN | $86.54 \pm 0.12$ | $61.76 \pm 0.14$ | $89.65 \pm 0.52$ | $92.70 \pm 0.20$ | $92.92 \pm 0.12$ | $96.18 \pm 0.07$ |
| GAT | $86.32 \pm 0.16$ | $64.47 \pm 0.18$ | $90.78 \pm 0.13$ | $93.87 \pm 0.11$ | $93.61 \pm 0.14$ | $96.17 \pm 0.08$ |
| APPNP | $88.43 \pm 0.15$ | $65.16 \pm 0.28$ | $90.18 \pm 0.17$ | $94.32 \pm 0.14$ | $94.49 \pm 0.07$ | $96.54 \pm 0.07$ |
| GPRGNN | $89.34 \pm 0.25$ | $67.12 \pm 0.31$ | $89.32 \pm 0.29$ | $94.49 \pm 0.14$ | $95.13 \pm 0.09$ | $96.85 \pm 0.08$ |
| GraphSAINT | $88.96 \pm 0.16$ | $67.85 \pm 0.21$ | $90.22 \pm 0.15$ | $91.72 \pm 0.13$ | $94.41 \pm 0.09$ | $96.43 \pm 0.05$ |
| PPRGo | $87.38 \pm 0.11$ | $63.54 \pm 0.25$ | $88.69 \pm 0.21$ | $93.61 \pm 0.12$ | $92.52 \pm 0.15$ | $95.51 \pm 0.08$ |
| GRAND+ | $88.64 \pm 0.09$ | $71.37 \pm 0.11$ | $88.74 \pm 0.11$ | $94.75 \pm 0.12$ | $93.92 \pm 0.08$ | $96.47 \pm 0.04$ |
| GT | $88.79 \pm 0.12$ | $61.05 \pm 0.38$ | $91.18 \pm 0.17$ | $94.74 \pm 0.13$ | $94.64 \pm 0.13$ | $97.05 \pm 0.05$ |
| Graphormer | OOM | OOM | OOM | $92.74 \pm 0.14$ | OOM | OOM |
| SAN | $88.22 \pm 0.15$ | $59.01 \pm 0.34$ | $89.83 \pm 0.16$ | $94.86 \pm 0.10$ | $94.51 \pm 0.15$ | OOM |
| GraphGPS | $88.94 \pm 0.16$ | $55.76 \pm 0.23$ | OOM | $95.06 \pm 0.13$ | $93.93 \pm 0.12$ | OOM |
| NAGphormer | $\mathbf{89.70 \pm 0.19}$ | $\mathbf{71.51 \pm 0.13}$ | $\mathbf{91.22 \pm 0.14}$ | $\mathbf{95.49 \pm 0.11}$ | $\mathbf{95.75 \pm 0.09}$ | $\mathbf{97.34 \pm 0.03}$ |

# 4 EXPERIMENTS

## 4.1 EXPERIMENTAL SETUP

Here we briefly introduce datasets and baselines in experiments. More details are in Appendix D.

**Datasets**. We conduct experiments on nine widely used datasets of various scales, including six small-scale datasets and three relatively large-scale datasets. For small-scale datasets, we adopt Pubmed, CoraFull, Computer, Photo, CS and Physics from the Deep Graph Library (DGL). We apply 60%/20%/20% train/val/test random splits for small-scale datasets. For large-scale datasets, we adopt AMiner-CS, Reddit and Amazon2M from (Feng et al., 2022). The splits of large-scale datasets are followed the settings from (Feng et al., 2022). Detailed information is provided in Appendix D.1.

**Baselines**. We compare NAGphormer with twelve advanced baselines, including: (I) four full-batch GNNs: GCN (Kipf & Welling, 2017), GAT (Veličković et al., 2018), APPNP (Klicpera et al., 2019) and GPRGNN (Chien et al., 2021); (II) three scalable GNNs: GraphSAINT (Zeng et al., 2020), PPRGo (Bojchevski et al., 2020) and GRAND+ (Feng et al., 2022); (III) five graph Transformers[1]: GT (Dwivedi & Bresson, 2020), SAN (Kreuzer et al., 2021), Graphormer (Ying et al., 2021), GraphGPS (Rampásek et al., 2022) and Gophormer (Zhao et al., 2021)[2].

## 4.2 COMPARISON ON SMALL-SCALE DATASETS

We conduct 10 trials with random seeds for each model and take the mean accuracy and standard deviation for comparison on small-scale datasets, and the results are reported in Table 1. From the experimental results, we can observe that NAGphormer outperforms the baselines consistently on all these datasets. For the superiority over GNN-based methods, it is because NAGphormer utilizes Hop2Token and the Transformer model to capture the semantic relevance of different hop neighbors overlooked in most GNNs, especially compared to APPNP and GPRGNN, which are two decoupled GCNs. Besides, the performance of NAGphormer also surpasses graph Transformer-based methods, indicating that leveraging the local information is beneficial for node classification. In particular, NAGphormer outperforms GT and SAN, which also introduce the eigenvectors of Laplacian matrix as the structural encoding into Transformers for learning the node representations, demonstrating the superiority of our proposed NAGphormer. Moreover, We observe that Graphormer, SAN, and GraphGPS suffer from the out-of-memory error even in some small graphs, further demonstrating the necessity of designing a scalable graph Transformer for large-scale graphs.

---

[1]Another recent graph Transformer, SAT (Chen et al., 2022), is not considered as it reports OOM even in our small-scale graphs.

[2]We compare the performance of our NAGphormer with Gophormer on the datasets reported in the original paper (Zhao et al., 2021) in Appendix E since the authors did not make their code available, and we implemented their algorithm but could not reproduce the same results as reported in their paper. Nevertheless, it still shows that our results are very competitive with their reported results.

Table 2: Comparison of all models in terms of mean accuracy $\pm$ stdev (%) on large-scale datasets. The best results appear in **bold**.

| Method | AMiner-CS | Reddit | Amazon2M |
|---|---|---|---|
| PPRGo | $49.07 \pm 0.19$ | $90.38 \pm 0.11$ | $66.12 \pm 0.59$ |
| GraphSAINT | $51.86 \pm 0.21$ | $92.35 \pm 0.08$ | $75.21 \pm 0.15$ |
| GRAND+ | $54.67 \pm 0.25$ | $92.81 \pm 0.03$ | $75.49 \pm 0.11$ |
| NAGphormer | $\mathbf{56.21 \pm 0.42}$ | $\mathbf{93.58 \pm 0.05}$ | $\mathbf{77.43 \pm 0.24}$ |

Table 3: The accuracy (%) with or without structural encoding.

| | Pubmed | CoraFull | CS | Computer | Photo | Physics | Aminer-CS | Reddit | Amazon2M |
|---|---|---|---|---|---|---|---|---|---|
| W/O-SE | 89.06 | 70.42 | 95.52 | 90.44 | 95.02 | 97.10 | 55.64 | 93.47 | 76.98 |
| With-SE | 89.70 | 71.51 | 95.75 | 91.22 | 95.49 | 97.34 | 56.21 | 93.58 | 77.43 |
| Gain | +0.64 | +1.09 | +0.23 | +0.78 | +0.47 | +0.24 | +0.57 | +0.11 | +0.45 |

### 4.3 COMPARISON ON LARGE-SCALE DATASETS

To verify the scalability of NAGphormer, we continue the comparison on three large-scale datasets. For the baselines, we only compare with three scalable GNNs, as existing graph Transformers can not work on such large-scale datasets due to their high computational cost. The results are summarized in Table 2. NAGphormer consistently outperforms the scalable GNNs on all datasets, indicating that NAGphormer can better preserve the local information of nodes and is capable of handling the node classification task in large graphs. The cost of training the model is reported in Appendix G, which demonstrates the efficiency of NAGphormer for handling large graphs.

### 4.4 ABLATION STUDY

To analyze the effectiveness of structural encoding and attention-based readout function, we perform a series of ablation studies on all datasets.

**Structural encoding**. We compare our proposed NAGphormer to its variant without the structural encoding module to measure the gain of structural encoding. The results are summarized in Table 3. We can observe that the gains of adding structural encoding vary in different datasets, since different graphs exhibit different topology structure. Therefore, the gain of structural encoding is sensitive to the structure of graphs. These results also indicate that introducing the structural encoding can improve the model performance for the node classification task.

**Attention-based readout function**. We conduct a comparative experiment between the proposed attention-based readout function ATT. (Equation 11) with previous readout functions, *i.e.*, SIN. and SUM.. The function of SIN. utilizes the corresponding representation of the node itself learned by the Transformer layer as the final output to predict labels. And SUM. can be regarded as aggregating all information of different hops equally. From Figure 2, we observe that ATT. outperforms other readout functions on small-scale datasets, indicating that aggregating information from different neighborhoods adaptively is beneficial to learn more expressive node representations, further improving the model performance on node classification. Due to the page limitation, the results on large-scale datasets which exhibit similar observations are provided in Appendix F.

### 4.5 PARAMETER STUDY

To further evaluate the performance of NAGphormer, we study the influence of two key parameters: the number of propagation steps $K$ and the number of Transformer layers $L$. Specifically, we perform experiments on AMiner-CS, Reddit and Amazon2M by setting different values of $K$ and $L$.

**On parameter $K$**. We fix $L = 1$ and vary the number of propagation steps $K$ in $\{4, 6, \cdots, 20\}$ Figure 3(a) reports the model performance. We can observe that the values of $K$ are different for each dataset to achieve the best performance since different networks exhibit different neighborhood structures. Besides, we can also observe that the model performance does not decline significantly even if $K$ is relatively large to 20. For instance, the performance on Reddit dataset changes slightly ($< 0.1\%$) with the increment of $K$, which indicates that learning the node representations from the

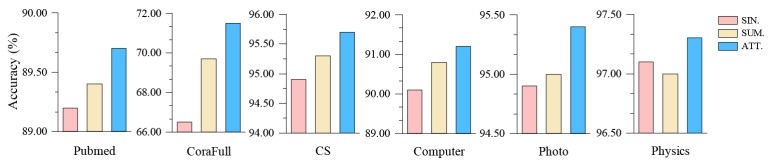

Figure 2: The performance of NAGphormer via different readout functions.

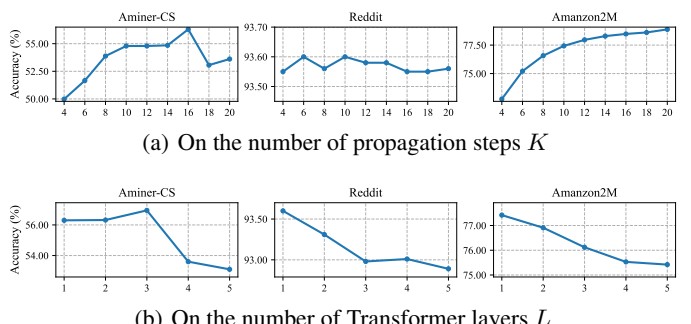

(a) On the number of propagation steps $K$

(b) On the number of Transformer layers $L$

Figure 3: Performance of NAGphormer on different parameters.

information of multi-hop neighborhoods via the self-attention mechanism and attention-based readout function can alleviate the impact of over-smoothing or over-squashing problems. In addition, the model performance changes differently on three datasets with the increment of $K$. The reason may be that these datasets are different types of networks and have diverse properties. This observation also indicates that neighborhood information on different types of networks has different effects on the model performance. In practice, we set $K = 16$ for AMiner-CS, and set $K = 10$ for others since the large propagation step will bring the high time cost of Hop2Token on Amanzon2M.

**On parameter** $L$. We fix the best value of $K$ and vary $L$ from 1 to 5 on each dataset. The results are shown in Figure 3(b). Generally speaking, a smaller $L$ can achieve a high accuracy while a larger $L$ degrades the performance of NAGphormer. Such a result can attribute to the fact that a larger $L$ is more likely to cause over-fitting. we set $L = 3$ for AMiner-CS, and set $L = 1$ for other datasets.

## 5 CONCLUSION

We propose NAGphormer, a novel and powerful graph Transformer for the node classification task. By utilizing a novel module Hop2Token to extract the features of different-hop neighborhoods and transform them into tokens, NAGphormer treats each node as a sequence composed of the corresponding neighborhood tokens. In this way, graph structural information of each node could be carefully preserved. Meanwhile, such a tokenized design enables NAGphormer to be trained in a mini-batch manner, enabling NAGphormer handle large-scale graphs. In addition, NAGphormer develops an attention-based readout function for learning the node representation from multi-hop neighborhoods adaptively, further boosting the model performance. We also provide theoretical analysis indicating that NAGphormer can learn more expressive node representations than the decoupled GCN. Experiments on various datasets from small to large demonstrate the superiority of NAGphormer over representative graph Transformers and Graph Neural Networks. Our tokenized design makes graph Transformers possible to handle large graphs. We wish our work inspire more works in this direction, and in our future work, we will generalize NAGphormer to other graph mining tasks, such as graph classification.

## ACKNOWLEDGMENTS

This work is supported by National Natural Science Foundation (U22B2017,62076105).

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

# A    RELATED WORK

## A.1    GRAPH NEURAL NETWORK

Graph Neural Network (GNN) has become a powerful technique for modeling the graph-structured data. Based on the message passing mechanism, GNN can learn the node representations from topology features and attribute features simultaneously. Typical GNNs, such as GCN (Kipf & Welling, 2017) and GAT (Veličković et al., 2018), leverage the features of immediate neighbors via different aggregation strategies to learn the node representations, exhibiting competitive performance on various graph mining tasks. However, typical GNNs obey the coupled design that binds the aggregation module and the feature transformation module in each GNN layer, leading to the over-smoothing (Chen et al., 2020a) and over-squashing issues (Alon & Yahav, 2021) on high-layer GNNs. Such a problem limits the model's ability to capture deep graph structural information. A reasonable solution is to decouple the aggregation and feature transformation modules in a GNN layer, treating them as independent modules (Klicpera et al., 2019; Wu et al., 2019; Chien et al., 2021), termed decoupled Graph Convolutional Network (decoupled GCN) (Dong et al., 2021). Decoupled GCN utilizes various propagation methods, such as personalized PageRank Klicpera et al. (2019) and random walk Wu et al. (2019), to aggregate features of multi-hop neighborhoods and further generate the node representations. Since the nonlinear activation functions between GNN layers are removed, decoupled GCN exhibits high computational efficiency and has become an advanced type of GNN in recent years. Besides the decoupled strategy, recent works (Yang et al., 2020; Lu et al., 2021; Huang et al., 2020; Sun et al., 2022) make efforts to address the over-smoothing and over-squashing issues by developing novel training tricks (Yang et al., 2020; Huang et al., 2020) or new graph neural network architectures (Lu et al., 2021; Sun et al., 2022). By introducing the carefully designed techniques, the impact of over-smoothing and over-squashing problems in GNNs could be well alleviated.

**Scalable GNN.** Most GNNs (Chen et al., 2020c; Jin et al., 2021; Kipf & Welling, 2017; Veličković et al., 2018) require the entire adjacency matrix as the input during training. In this way, when applying to large-scale graphs, the cost of training is too high to afford. There are two categories of strategies for generalizing GNN to large-scale graphs:

(I) The node sampling strategy (Hamilton et al., 2017; Chiang et al., 2019; Zou et al., 2019; Zeng et al., 2020) that sample partial nodes from the whole graph via different methods, such as random sampling from neighbors (Hamilton et al., 2017) and sampling from GNN layers (Zou et al., 2019), to reduce the size of nodes for model training.

(II) The approximation propagation (Chen et al., 2020b; Feng et al., 2022; Bojchevski et al., 2020) that accelerates the propagation operation via several approximation methods, such as approximate PageRank (Bojchevski et al., 2020) and sub-matrix approximation (Feng et al., 2022).

However, by designing various sampling-based or approximation-based methods to reduce the training cost, these models will inevitably lead to information loss and somehow restrict their performance on large-scale networks.

## A.2    GRAPH TRANSFORMER

In existing graph Transformers, there are mainly three strategies to incorporate graph structural information into the Transformer architecture so as to learn the node representations:

(I) Extracting the positional embedding from graph structure. Dwivedi *et al*. (Dwivedi & Bresson, 2020) utilize Laplacian eigenvectors to represent positional encodings of the original Transformer and fuse them with the raw attributes of nodes as the input. Derived from (Dwivedi & Bresson, 2020), Devin *et al*. (Kreuzer et al., 2021) leverage the full spectrum of Laplacian matrix to learn the positional encodings.

(II) Combining GNN and Transformer. In addition to representing structural information by the eigenvectors, Wu *et al*. (Jain et al., 2021) regard GNNs as an auxiliary module to extract fixed local structural information of nodes and further feed them into the Transformer to learn long-range pairwise relationships. Chen *et al*. (Chen et al., 2022) utilize a GNN model as the structure extractor to learn different types of structural information, such as $k-$subtree and $k-$subgraph, to capture the structure similarity of node pairs via the self-attention mechanism. Rampášek *et al*. (Rampásek et al.,

2022) develop a hybrid layer that contains a GNN layer and a self-attention layer to capture both local and global information.

(III) Integrating the graph structural bias into the self-attention matrix. There are several efforts to transform various graph structure features into attention biases and integrate them into the self-attention matrix to enable the Transformer to capture graph structural information. Ying *et al.* (Ying et al., 2021) propose a spatial encoding method that models the structural similarity of node pairs based on the length of their shortest path. Zhao *et al.* (Zhao et al., 2021) propose a proximity-enhanced attention matrix by considering the relationship of node pairs in different neighborhoods. Besides, by modeling edge features in chemical and molecular graphs, Dwivedi *et al.* (Dwivedi & Bresson, 2020) extend graph Transformers to edge feature representation by injecting them into the self-attention module of Transformers. Hussain *et al.* (Hussain et al., 2022) utilize the edge features to strengthen the expressiveness of the attention matrix. Wu *et al.* (Wu et al., 2022) introduce the topology structural information as the relational bias to strengthen the original attention matrix.

Nevertheless, except GraphGPS (Rampásek et al., 2022) and Nodeformer (Wu et al., 2022) whose complexities are linear to the number of nodes and edges, the aforementioned methods adopt the fully-connected attention mechanism upon all the node pairs, in which the spatial complexity is quadratic with the number of nodes. Such high complexity makes these methods hard to directly handle graph mining tasks on large-scale networks with millions of nodes and edges. A recent work (Zhao et al., 2021) samples several ego-graphs of each node and then utilizes Transformer to learn the node representations on these ego-graphs so as to reduce the computational cost of model training. However, the sampling process is still time-consuming in large graphs. Moreover, the sampled ego-graphs only contain limited neighborhood information due to the fixed and small sampled graph size for all nodes, which is insufficient to learn the informative node representations.

## B  COMPLEXITY ANALYSIS OF NAGPHORMER

This section provides the complexity analysis of NAGphormer on time and space.

**Time complexity.** The time complexity of NAGphormer mainly depends on the self-attention module of the Transformer. So the computational complexity of NAGphormer is $O(n(K+1)^2 d)$, where $n$ denotes the number of nodes, $K$ denotes the number of hops and $d$ the dimension of parameter matrix (*i.e.*, feature vector).

**Space complexity.** The space complexity is based on the number of model parameters and the outputs of each layer. The first part is mainly on the Transformer layer $O(d^2 L)$, where $L$ is the number of Transformer layers. The second part is on the attention matrix and the hidden node representations, $O(b(K+1)^2 + b(K+1)d)$, where $b$ denotes the batch size. Thus, the total space complexity is $O(b(K+1)^2 + b(K+1)d + d^2 L)$.

## C  PROOF OF **FACT 1**

Here we provide the detailed proof for **Fact 1**.

**Fact 1.** *From the perspective of the output node representations of Hop2Token, we can regard the decoupled GCN as applying a self-attention mechanism with a fixed attention matrix $\mathbf{S} \in \mathbb{R}^{(K+1) \times (K+1)}$, where $\mathbf{S}_{K,k} = \beta_k$ ($k \in \{0, ..., K\}$) and other elements are all zeroes.*

*Proof.* First, both Hop2Token and decouple GCN utilize the same propagation process to obtain the information of different-hop neighborhoods. So we use the same symbol $\mathbf{H}_i^{(k)} \in \mathbb{R}^{1 \times d}$ to represent the neighborhood information of node $i$ at propagation step $k$ for brevity.

For an arbitrary node $i$, each element $\mathbf{Z}_{i,m}(m \in \{1, ..., d\})$ of the output representation $\mathbf{Z}_i \in \mathbb{R}^{1 \times d}$ learned by the decoupled GCN according to Equation 2 is calculated as:

$$\mathbf{Z}_{i,m} = \sum_{k=0}^{K} \beta_k \mathbf{H}_{i,m}^{(k)}. \tag{13}$$

Table 4: Statistics on datasets.

| Dataset | # Nodes | # Edges | # Features | # Classes |
|---|---|---|---|---|
| Pubmed | 19,717 | 44,324 | 500 | 3 |
| CoraFull | 19,793 | 126,842 | 8,710 | 70 |
| Computer | 13,752 | 491,722 | 767 | 10 |
| Photo | 7,650 | 238,163 | 745 | 8 |
| CS | 18,333 | 163,788 | 6,805 | 15 |
| Physics | 34,493 | 495,924 | 8,415 | 5 |
| AMiner-CS | 593,486 | 6,217,004 | 100 | 18 |
| Reddit | 232,965 | 11,606,919 | 602 | 41 |
| Amazon2M | 2,449,029 | 61,859,140 | 100 | 47 |

On the other hand, the output $\mathbf{X}_i \in \mathbb{R}^{(K+1) \times d}$ of Hop2Token in the matrix form for node $i$ is described as:

$$\mathbf{X}_i = \begin{bmatrix} \mathbf{H}_{i,0}^{(0)} & \mathbf{H}_{i,1}^{(0)} & \cdots & \mathbf{H}_{i,d}^{(0)} \\ \mathbf{H}_{i,0}^{(1)} & \mathbf{H}_{i,1}^{(1)} & \cdots & \mathbf{H}_{i,d}^{(1)} \\ \vdots & \vdots & \ddots & \vdots \\ \mathbf{H}_{i,0}^{(K)} & \mathbf{H}_{i,1}^{(K)} & \cdots & \mathbf{H}_{i,d}^{(K)} \end{bmatrix}. \tag{14}$$

Suppose we have the following attention matrix $\mathbf{S} \in \mathbb{R}^{(K+1) \times (K+1)}$:

$$\mathbf{S} = \begin{bmatrix} 0 & 0 & \cdots & 0 \\ 0 & 0 & \cdots & 0 \\ \vdots & \vdots & \ddots & \vdots \\ \beta_0 & \beta_1 & \cdots & \beta_K \end{bmatrix}. \tag{15}$$

Following Equation 4, the output matrix $\mathbf{T} \in \mathbb{R}^{(K+1) \times d}$ learned by the self-attention mechanism can be described as:

$$\mathbf{T} = \mathbf{S}\mathbf{X}_i = \begin{bmatrix} 0 & 0 & \cdots & 0 \\ 0 & 0 & \cdots & 0 \\ \vdots & \vdots & \ddots & \vdots \\ \gamma_0 & \gamma_1 & \cdots & \gamma_d \end{bmatrix}, \tag{16}$$

where $\gamma_m = \sum_{k=0}^{K} \beta_k \mathbf{H}_{i,m}^{(k)} (m \in \{1, ..., d\})$.

Further, we can obtain each element $\mathbf{T}_m^{final} (m \in \{1, ..., d\})$ of the final representation $\mathbf{T}^{final} \in \mathbb{R}^{1 \times d}$ of node $i$ by using a summation readout function:

$$\mathbf{T}_m^{final} = \sum_{k=0}^{K} \mathbf{T}_{k,m} = (0 + 0 + \cdots + \gamma_m) = \sum_{k=0}^{K} \beta_k \mathbf{H}_{i,m}^{(k)} = \mathbf{Z}_{i,m}. \tag{17}$$

Finally, we can obtain **Fact 1**.

# D EXPERIMENTAL DETAILS

## D.1 DATASET DESCRIPTION

Here we provide detailed descriptions for each dataset. Pubmed (Namata et al., 2012), Cora-Full (Shchur et al., 2018) and AMiner-CS (Feng et al., 2020) are citation networks in which nodes represent papers and edges represent citations. Computer (Shchur et al., 2018), Photo (Shchur et al., 2018), Amazon2M (Chiang et al., 2019) are co-purchase networks, where nodes indicate goods and edges indicate that the two connected goods are frequently bought together. CS (Shchur et al., 2018) and Physics (Shchur et al., 2018) are co-authorship networks, where nodes denote authors and edges

Table 5: Further comparison of NAGphormer and Gophormer. The best results appear in **bold**.

| | Cora | Citeseer | Pumbed | Blogcatalog | DBLP | Flickr |
|---|---|---|---|---|---|---|
| Gophormer | 87.85 | **80.23** | 89.40 | 96.03 | 85.20 | 91.51 |
| NAGphormer | **88.15** | 80.12 | **89.70** | **96.73** | **85.95** | **91.80** |

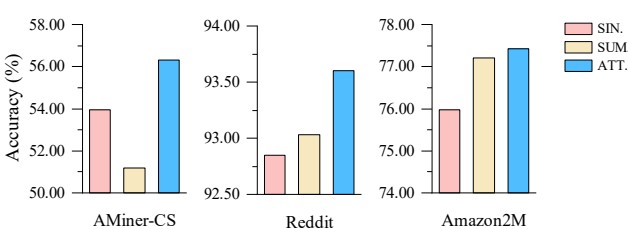

Figure 4: The performance of different readout functions on large-scale datasets.

represent that authors have co-authored at least one paper. Reddit (Hamilton et al., 2017) is a social network where nodes represent posts and edges denote that the same user has commented on the two connected posts. The statistics on datasets are reported in Table 4.

### D.2 IMPLEMENTATION DETAILS

Referring to the recommended settings in the official implementations, we perform hyperparameter tuning for each baseline. For the model configuration of NAGphormer, we try the number of Transformer layers in $\{1, 2, ..., 5\}$, the hidden dimension in $\{128, 256, 512\}$, and the propagation steps in $\{2, 3, ..., 20\}$. Parameters are optimized with the AdamW (Loshchilov & Hutter, 2019) optimizer, using a learning rate of in $\{1e-3, 5e-3, 1e-4\}$ and weight decay of $\{1e-4, 5e-4, 1e-5\}$. We also search the dropout rate in $\{0.1, 0.3, 0.5\}$. The batch size is set to 2000. The training process is early stopped within 50 epochs. All experiments are conducted on a Linux server with 1 I9-9900k CPU, 1 RTX 2080TI GPU and 64G RAM.

### E  FURTHER COMPARISON WITH GOPHORMER

Gophormer (Zhao et al., 2021) is a recent work on arXiv that generalizes the Transformer on graphs for the node classification task. Since the code or model of Gophormer is not available, for the comparison of NAGphormer and Gophormer, we run NAGphormer on the datasets used in the original paper (Zhao et al., 2021) with the same ratio of random splits. Our results in terms of accuracy (%) are reported in Table 5. For Gophormer, we use their reported results (Zhao et al., 2021). We can observe that NAGphormer outperforms Gophormer on all datasets except Citeseer even on their data using their reported results, demonstrating that NAGphormer is more potent than Gophormer on the node classification task.

### F  ABLATION STUDY OF READOUT FUNCTION ON LARGE-SCALE DATASETS

Figure 4 exhibits the performance of NAGphormer via different readout functions on large-scale datasets. The results show that our proposed attention-based readout function consistently outperforms others (SIN. and SUM.) on three large-scale datasets, demonstrating that learning the node representations from multi-hop neighborhoods adaptively can improve the model performance for the node classification task.

Table 6: The training cost on large-scale graphs in terms of GPU memory (MB) and running time (s).

| | Aminer-CS | | Reddit | | Amazon2M | |
|---|---|---|---|---|---|---|
| | Memory (MB) | Time (s) | Memory (MB) | Time (s) | Memory (MB) | Time (s) |
| GraphSAINT | 1,641 | 23.67 | 2,565 | 43.15 | 5,317 | 334.08 |
| PPRGo | 1,075 | 14.21 | 1,093 | 35.73 | 1,097 | 152.62 |
| GRAND+ | 1,091 | 21.41 | 1,213 | 197.97 | 1,123 | 207.85 |
| NAGphormer | 1,827 | 19.87 | 1,925 | 20.72 | 2,035 | 58.66 |

## G  EFFICIENCY EXPERIMENTS ON LARGE-SCALE GRAPHS

In this section, we validate the efficiency of NAGphormer on large-scale graphs. Specifically, we compare the training cost in terms of running time (s) and GPU memory (MB) of NAGphormer and three scalable GNNs, PPRGo, GraphSAINT and GRAND+. For scalable GNNs, We adopt the official implements on Github. However, all methods contain diverse pre-processing steps built on different programming language frameworks, such as approximate matrix-calculation based on C++ framework in GRAND+. To ensure the fair comparison, we report the running time cost including the training stage and inference stage since these stages of all models are based on Pytorch framework. The results are summarized in Table 6.

From the experimental results, we can observe that NAGphormer shows high efficiency when dealing with large graphs. For instance, on Amazon2M which contains two million nodes and 60 million edges, NAGphormer achieves almost $3\times$ acceleration compared with the second fastest model PPRGo. The reason is that the time complexity of NAGphormer is mainly depended on the number of nodes and has nothing with the number of edges, while the time consumption of other methods is related to the number of edges and nodes since these methods involve the propagation operation during the training and inference stages. As for the GPU memory cost, since NAGphormer utilizes the mini-batch training, the GPU memory cost is determined by the batch size. Hence, the GPU memory cost of NAGphormer is affordable by choosing a proper batch size even on large-scale graphs.

