# OpenReview forum: "NAGphormer: A Tokenized Graph Transformer for Node Classification in Large Graphs"
_ICLR.cc/2023/Conference — ICLR 2023 poster_

### Official Review · Reviewer_b2CU · 2022-10-23

**Confidence:** 3
**Clarity, Quality, Novelty And Reproducibility:** 1. The paper is written clearly.
2. T…
**Correctness:** 4
**Technical Novelty And Significance:** 3
**Empirical Novelty And Significance:** 3
**Recommendation:** 8

**Strength And Weaknesses:**

Strengths:

1. With node-wise sequences as inputs, mini-batch can be used, and training becomes more scalable.
2. Extensive experiments are conducted to compare NAGphormer with many baseline models in small to large-scale tasks. Additionally, an ablation study on the effect of different readout functions and propagation steps is performed.
3. The empirical performance of NAGphormer is outstanding.


Weaknesses:

1. The authors propose that with Hop2Token, NAGphormer is more expressive than decoupled GCN family. However, in the experimental section, no decoupled GCN models are compared. Therefore, it will be more persuasive if the authors can add some of the decoupled GCN as baselines.
2. Given the good performance of NAGphormer, from the ablation study over the depth of the model, it seems that NAGphormer also suffers from oversmoothing or oversquashing problems since as the depth increases, the performance deteriorates.


**Summary Of The Paper:**

This paper proposes a new neighborhood aggregation method based on the number of hops. With this method, a graph transformer is developed with node-wise input sequences. Moreover, a self-attention readout function is applied to integrate the representations learned from the transformer module.

**Summary Of The Review:**

The idea of node-wise sequence input is novel, and the proposed model is competitive compared with current SOTAs.

---

> ### Author Response · Authors · 2022-11-15
> **Response to reviewer b2CU**
>
> Thank you for your kind and thoughtful reviews. We address your concerns as follows.
>
> >Q1. "The authors propose that with Hop2Token, NAGphormer is more expressive than decoupled GCN family. However, in the experimental section, no decoupled GCN models are compared. Therefore, it will be more persuasive if the authors can add some of the decoupled GCN as baselines."
>
> A1. Thank you for your suggestion. In the original paper, we select two powerful decoupled GCNs, APPNP [1] and GPRGNN [2], as baselines. The experimental results are reported in Table 1. We have highlighted that the above two methods are decoupled GCNs in the revised version, which can clearly demonstrate the superiority of our approach. Thank you.
>
> >Q2. "Given the good performance of NAGphormer, from the ablation study over the depth of the model, it seems that NAGphormer also suffers from oversmoothing or oversquashing problems since as the depth increases, the performance deteriorates."
>
> A2. The token corresponding to the large hop of the neighborhood could suffer from over-smoothing or over-squashing problems since we utilize the propagation operation to calculate the token features. Nevertheless, NAGphormer can adaptively learn the final node representation from all tokens via the self-attention mechanism and attention-based readout function, which can mitigate the impact of such problems.
>
> In addition, we vary the propagation step $k$ in a larger range {$4,\cdots,20\$} compared to the original settings and report the model performance. The experimental results are shown in the following table.
>
> |         | 4     | 6     | 8     | 10    | 12    | 14    | 16    | 18    | 20    |
> |---------|-------|-------|-------|-------|-------|-------|-------|-------|-------|
> | Aminer  | 49.98 | 51.66 | 53.88 | 54.79 | 54.79 | 54.85 | 56.3  | 53.06 | 53.61 |
> | Reddit  | 93.55 | 93.59 | 93.56 | 93.60 | 93.58 | 93.58 | 93.55 | 93.55 | 93.56 |
> | Amanzon | 72.76 | 75.21 | 76.57 | 77.43 | 77.96 | 78.29 | 78.48 | 78.61 | 78.88 |
>
> We can observe that the model performance does not decline significantly even if setting a large value of $k$. For instance, the performance on Reddit dataset changes slightly (<0.1%) with the increase of $k$. This phenomenon indicates that NAGphormer can alleviate the influence of over-smoothing or over-squashing problems.
>
> In addition, the model performance changes differently on three graphs with the increment of $k$. The reason may be that these three datasets are different types of networks and have diverse properties. Aminer-CS is a citation network, Reddit is a social network and Amazon2M is a co-purchase network. This observation also indicates that neighborhood information on different types of networks has different effects on the model performance.
>
> We also add the discussion in the revision paper in Section 4.5 about the parameter $k$.
>
> [1] Johannes Klicpera, et al. Predict then Propagate: Graph Neural Networks Meet Personalized Pagerank. ICLR 2019.
>
> [2] Eli Chien, et al. Adaptive Universal Generalized Pagerank Graph Neural Network. ICLR 2021.

---

### Official Review · Reviewer_zQFG · 2022-10-24

**Confidence:** 3
**Correctness:** 3
**Technical Novelty And Significance:** 3
**Empirical Novelty And Significance:** 2
**Recommendation:** 5

**Clarity, Quality, Novelty And Reproducibility:**

The paper is clearly written. To the best of my knowledge, Hop2Token is novel, although heavily related to the decoupled GCN and APPNP lines of work. The NAGphormer is less novel beyond Hop2Token and seems mainly serves as a way to adapt a standard transformer to Hop2Token (e.g. structural encoding has already been studied in previous works).

**Strength And Weaknesses:**

Strength
- The proposed Hop2Token method is novel, simple, and can be efficiently implemented.
- NAGphormer demonstrates empirical gains over thorough experiments and ablations.
- The paper is clearly written and easy to follow.

Weaknesses
- Several important previous works are not mentioned and compared against, most notably "Recipe for a General, Powerful, Scalable Graph Transformer", which proposes reducing computation costs using a linear transformer. Other recent graph transformers like K-Subgraph SAT and EGT should also be compared if the authors wish to support the claim "consistently outperforms existing graph Transformers and mainstream GNNs".
- The expressivity result against decoupled GCN does not seem very strong, since decoupled GCN is very simple and limited itself (probably by WL1 test, also https://arxiv.org/pdf/2010.12408.pdf shows that it can be seen as two times label propagations.)
- Hop2Token fundamentally tries to compress an exponentially more amount of nodes information to a single token as the hop k increases. This is known to suffer from over-squashing issue. Although this may not be shown in the experimented datasets (maybe due to high homophily), this is a concern conceptually.


**Summary Of The Paper:**

This paper proposes Hop2Token to scale a graph transformer to larger graphs by using multi-hop aggregated features of each node as the input tokens to graph transformer, instead of using all nodes as the tokens. The overall transformer using Hop2Token is called NAGphormer, which also uses structural encoding and attention-based readout function. The method is tested on both small and large node classification datasets and demonstrates empirical gains.

**Summary Of The Review:**

Overall, the proposed Hop2Token seems a conceptually interesting and promising way to enable mini-batch training over larger graphs. However, several important baselines are missing and the method in its current form could suffer from severe over-squashing problems.

---

> ### Author Response · Authors · 2022-11-15
> **Response to reviewer zQFG**
>
> Thank you for the detailed comments and valuable questions. We provide details to clarify your major concerns.
>
> >Q1. "Several important previous works are not mentioned and compared against, most notably "Recipe for a General, Powerful, Scalable Graph Transformer", which proposes reducing computation costs using a linear transformer. Other recent graph transformers like K-Subgraph SAT and EGT should also be compared if the authors wish to support the claim "consistently outperforms existing graph Transformers and mainstream GNNs"."
>
> A1. Per your suggestion, we conduct additional experiments to evaluate the performance of GraphGPS [1] and SAT [2]. Since datasets adopted in this paper do not contain edge features, we ignore EGT [3], whose key idea is to utilize the edge channels to capture the structural information.
> We adopt the official implements of GraphGPS and SAT on Github. As SAT meets the Out-of-Memory issue on all small-scale datasets which contain at least seven thousand nodes, we only report the performance of GraphGPS.
> | 　         | Pubmed | Corafull | Computer | Photo | CS    | Physics |
> |------------|--------|----------|----------|-------|-------|---------|
> | GraphGPS   | 88.94  | 55.76    | OOM      | 95.06 | 93.93 | OOM     |
> | NAGphormer | 89.70   | 71.51    | 91.22    | 95.49 | 95.75 | 97.34   |
>
> “OOM” indicates the Out-of-Memory problem. Since the complexity of GraphGPS is linear to the number of nodes and edges, it suffers from the huge memory overhead on large and dense graphs. We can observe that NAGphormer consistently surpasses GraphGPS on all datasets, which could support the claim.
>
> We have added the above experimental results in Table 1 of the revised version.
> We also have updated the section of Related Work to introduce GraphGPS and EGT.
>
> >Q2. "The expressivity result against decoupled GCN does not seem very strong, since decoupled GCN is very simple and limited itself (probably by WL1 test, also https://arxiv.org/pdf/2010.12408.pdf shows that it can be seen as two times label propagations.)"
>
> A2. As you mentioned, Dong et al. [4] only analyze early decoupled GCNs, such as APPNP [5] and SGC [6]. However, the advanced decoupled GCN, GPRGNN [7], has theoretically proved that it can learn from diverse label patterns (homophilic and heterophilic) and alleviate the impact of over-smoothing problem. In addition, they have shown promising performance on the node classification tasks over regular GCNs [5, 7]. Hence, we can regard decoupled GCN as a strong method for node classification tasks.
>
> In this paper, we focus on the node classification task. Compared with decoupled GCNs, our proposed NAGphormer can learn more informative node representations from multi-hop neighborhoods and show superior performance on various benchmark datasets, demonstrating the superiority of NAGphormer.
>
> >Q3. "Hop2Token fundamentally tries to compress an exponentially more amount of nodes information to a single token as the hop k increases. This is known to suffer from over-squashing issue. Although this may not be shown in the experimented datasets (maybe due to high homophily), this is a concern conceptually."
>
> A3. Thank you for your attention on this point. The token corresponding to the large hop neighborhood could suffer from the over-squashing issue. However, our final node representation is derived from all tokens via the self-attention mechanism and attention-based readout function, which enables the model to adaptively learn node representations from multi-hop neighborhoods, thus could alleviate these issues significantly. We also add the discussion in the revision version in Section 4.5 about the parameter $k$.
>
> [1] Ladislav Rampášek, et al. Recipe for a General, Powerful, Scalable Graph Transformer. NeurIPS 2022.
>
> [2] Dexiong Chen, et al. Structure-Aware Transformer for Graph Representation Learning. ICML 2022.
>
> [3] Md Shamim Hussain, et al. Global Self-Attention as a Replacement for Graph Convolution. KDD 2022.
>
> [4] Hande Dong, et al. On the Equivalence of Decoupled Graph Convolution Network and Label Propagation. WWW2021.
>
> [5] Johannes Klicpera, et al. Predict then Propagate: Graph Neural Networks Meet Personalized Pagerank. ICLR 2019.
>
> [6] Felix Wu, et al. Simplifying Graph Convolutional Networks. ICML 2019.
>
> [7] Eli Chien, et al. Adaptive Universal Generalized Pagerank Graph Neural Network. ICLR 2021.

---

### Official Review · Reviewer_gZ4g · 2022-10-24

**Confidence:** 4
**Correctness:** 4
**Technical Novelty And Significance:** 3
**Empirical Novelty And Significance:** 3
**Recommendation:** 8

**Clarity, Quality, Novelty And Reproducibility:**

- Clarity: easy to follow

- Quality: seem to be sound

- Novelty: fair

- Reproducibility: provided source code facilitates good reproducibility


**Strength And Weaknesses:**

Strengths:

- Different from existing graph transformer methods, this work treats different hops of neighborhood representations as tokens. This can make the proposed method scale to large graphs.

- The paper is written and organized well, and it is easy to follow.

- The provided source code facilitates good reproducibility of this work.

- The proposed method seems to have a promising performance in experiments.

Weaknesses:

- The authors argue that existing message passing-based GNNs have limitations such as over-smoothing and over-squashing. However, recently, there emerges a number of works e.g. [1,2,3,4,5] that address these limitations.

- The experiments are not extensive. Specifically, the running time cost and the memory cost of all the methods should be compared to better show the efficiency of the proposed method, since the efficiency of this work is highlighted throughout the text.

- This paper repeats itself in some pieces of text.

Refs:

[1] Huang, W., Rong, Y., Xu, T., Sun, F., & Huang, J. (2020). Tackling over-smoothing for general graph convolutional networks. arXiv preprint arXiv:2008.09864.

[2] Lu, W., Zhan, Y., Guan, Z., Liu, L., Yu, B., Zhao, W., ... & Tao, D. (2021). SkipNode: On alleviating over-smoothing for deep graph convolutional networks. arXiv preprint arXiv:2112.11628.

[3]	Yang, C., Wang, R., Yao, S., Liu, S., & Abdelzaher, T. (2020). Revisiting over-smoothing in deep GCNs. arXiv preprint arXiv:2003.13663.

[4] Huang, W., Rong, Y., Xu, T., Sun, F., & Huang, J. (2020). Tackling over-smoothing for general graph convolutional networks. arXiv preprint arXiv:2008.09864.

[5] Sun, Q., Li, J., Yuan, H., Fu, X., Peng, H., Ji, C., ... & Yu, P. S. (2022). Position-aware Structure Learning for Graph Topology-imbalance by Relieving Under-reaching and Over-squashing. arXiv preprint arXiv:2208.08302.

**Summary Of The Paper:**

This paper proposes a Graph Transformer architecture. It aggregates the neighborhood features from different hops and treat them as a sequence of token vectors for Transformer.

**Summary Of The Review:**

The idea is good. Although some necessary experiments are missing, I encourage the authors to supplement these experiments during the discussion phase, to well support their arguments.

---

> ### Author Response · Authors · 2022-11-15
> **Response to reviewer g24g**
>
> Many thanks for the positive evaluation and insightful questions that help us improve this work. We provide the following detailed responses to your questions.
>
> >Q1. "The authors argue that existing message passing-based GNNs have limitations such as over-smoothing and over-squashing. However, recently, there emerges a number of works e.g. [1,2,3,4] that address these limitations."
>
> A1. Thanks for your valuable comments. We have supplemented the recent works [1,2,3,4] addressing over-smoothing and over-squashing problems of GNNs in the related work section, and we also add some description in the introduction. Though these efforts have been devoted to alleviate the impact of over-smoothing and over-squashing problems, the negative influence of the inherent limitations cannot be eliminated completely.
>
> >Q2. "The experiments are not extensive. Specifically, the running time cost and the memory cost of all the methods should be compared to better show the efficiency of the proposed method, since the efficiency of this work is highlighted throughout the text."
>
> A2. Thank you for your helpful suggestion. To validate the efficiency, we report the running time and GPU memory cost of NAGphormer on three large-scale graphs, compared with three scalable GNNs, PPRGo, GraphSAINT and GRAND+, mentioned in the paper. We adopt their official implements on Github. However, all methods contain diverse pre-processing steps built on different programming language frameworks, such as approximate matrix calculation based on the C++ framework in GRAND+, which could affect the fairness of the efficiency experiment. Hence, we report the running time cost involving the model’s training stage and inference stage since these stages of all models are implemented by Pytorch. The experimental results, including GPU memory cost (MB) and running time cost (s), are summarized in the following table.
>
> |            | Aminer-CS |       | Reddit |        | Amazon2M |        |
> |------------|:---------:|:-----:|:------:|:------:|:--------:|:------:|
> |            |    Memory(MB) |  Time(s) | Memory(MB) |  Time(s) |   Memory(MB) |  Time(s) |
> | GraphSAINT |      1,641 | 23.67 |   2,565 |  43.15 |     5,317 | 334.08 |
> | PPRGo      |      1,075 | 14.21 |   1,093 |  35.73 |     1,097 | 152.62 |
> | GRAND+     |      1,091 | 21.41 |   1,213 | 197.97 |     1,123 | 207.85 |
> | NAGphormer |      1,827 | 19.87 |   1,925 |  20.72 |     2,035 |  58.66 |
>
> The results show that NAGphormer exhibits high efficiency when dealing with large graphs. For instance, on Amazon2M, which contains two million nodes and 60 million edges, NAGphormer achieves almost 3$\times$ acceleration compared with the second fastest model PPRGo. The reason is that the time complexity of NAGphormer mainly depends on the number of nodes, while the time cost of other methods is related to the number of edges and nodes since these methods involve the propagation operation during the training and inference stages.
>
> As for the GPU memory cost, since NAGphormer utilizes the mini-batch training, the GPU memory cost is determined by the batch size. Hence, the GPU memory cost of NAGphormer is affordable by choosing a proper batch size, even on large-scale graphs.
> We have added the above experimental results in Appendix G in the revised version.
>
> >Q3. "This paper repeats itself in some pieces of text."
>
> A3. We appreciate your attention to detail on this point. We have carefully checked the paper and reorganized the repeated sentences in the revised version.
>
> [1]. Qingyun Sun, et al. Position-aware Structure Learning for Graph Topology-imbalance by Relieving Under-reaching and Over-squashing. CIKM 2022.
>
> [2]. Wenbing Huang, et al. Tackling over-smoothing for general graph convolutional networks. arXiv 2020.
>
> [3]. Weigang Lu, et al. SkipNode: On alleviating over-smoothing for deep graph convolutional networks. arXiv 2021.
>
> [4]. Chaoqi Yang, et al. Revisiting over-smoothing in deep GCNs. arXiv 2020.

---

> > ### Comment · Reviewer_gZ4g · 2022-11-15
> > **The authors have addressed my concerns**
> >
> > Thank the authors for addressing my concerns. I have raised my recommendation scores.

---

> > > ### Author Response · Authors · 2022-11-15
> > > **Many thanks**
> > >
> > > Many thanks for your positive feedback that greatly encourages us.

---

### Official Review · Reviewer_wvBc · 2022-10-24

**Confidence:** 4
**Correctness:** 3
**Technical Novelty And Significance:** 2
**Empirical Novelty And Significance:** 2
**Recommendation:** 5

**Clarity, Quality, Novelty And Reproducibility:**

* Overall, the paper reads well, and the architecture is clearly presented with figures.
* Since the method is simple, it doesn't seem to have a problem with reproducibility.
* Novelty is somewhat limited. But processing the multiple tokens for a node separately and merging them in the attention-based readout layer is interesting. To prove the value of this architecture, the authors need to provide ablation studies or additional analyses.
* The authors claim that the proposed method is theoretically superior to Decoupled GCN. But decoupled GCN is not clearly introduced.
Although the proposed method exhibits empirically strong performance, technical contributions and novelty are limited.

**Strength And Weaknesses:**

Strengths:

* The proposed method shows competitive performance on many benchmark datasets.
* The multi-view/multi-scale representation of a node (a sequence of aggregated node features) is an interesting idea. The late fusion of the node features is new. However,  the attention-based readout can be viewed as deformable convolution with an adaptive receptive field since it varies neighborhood from 0-hop to K-hops. From this perspective, the authors may want to compare the proposed method with similar approaches in the literature.

Weaknesses:

* Unlike the authors' presentation, the model does not utilize order information at all. Indeed, the model can be viewed as an ensemble model with features from different layers of a GNN while sharing the transformer's parameters.
* One of the main contributions is "a sequence of tokens" for a node. But no ablation study is provided. For instance, simple feature concatenation is one naïve approach. Compared to straightforward construction, the authors should show the performance gain of the sequence of token node representation.
* Mini-batch training for large-scale graphs has already been studied in the literature. Unless the authors present a new technique, which significantly improves scalability, It is hard to consider the improved scalability as the contribution of this paper.

**Summary Of The Paper:**

This paper proposes a new graph Transformer architecture that contains a sequence of tokens obtained by aggregating neighbors' features. It shows improved expressive power with information from multi-hop neighborhoods. So, the model treats a node as a sequence of tokens. The authors claim that the model has better scalability due to its mini-bash training. The authors theoretically analyze that the proposed method has more expressive power compared to decoupled GCN.

**Summary Of The Review:**

The proposed method shows a significant performance gain. Also, the multi-hop representations for each node and merging transformed node features in the attention-based readout layer are interesting ideas. However, the authors did not provide experimental results to justify their architecture. Lastly, scalability is improved by well-known mini-batch training. Overall, this paper has limited novelty and a few missing analyses.

---

> ### Author Response · Authors · 2022-11-15
> **Response to reviewer wvBc**
>
> We appreciate the reviewer for providing valuable feedback and comments on our paper. Following are our detailed responses to your questions.
> > Q1. "Unlike the authors' presentation, the model does not utilize order information at all. Indeed, the model can be viewed as an ensemble model with features from different layers of a GNN while sharing the transformer's parameters."
>
> A1. In NAGphormer, the order information is obtained by calculating the eigenvectors of nodes in each hop neighborhood, which could be regarded as the position encoding of different hop neighborhoods. The detailed implementation is introduced in Section 3.3, structural encoding.
>
> Moreover, the feature matrix $\mathbf{H}^k$ learned by the $k$-th GNN layer is calculated through $k$ linear layers and nonlinear activation functions, which is significantly different from our approach since the proposed Hop2Token is a non-parametric method.
>
> >Q2. "One of the main contributions is "a sequence of tokens" for a node. But no ablation study is provided. For instance, simple feature concatenation is one naïve approach. Compared to straightforward construction, the authors should show the performance gain of the sequence of token node representation."
>
> A2. Thank you for pointing it out. For your suggestions, we concatenate the features of multi-hop neighborhoods generated by Hop2Token for each node, resulting in a new feature matrix as the input of the model. Since the attention-based readout function and mini-batch training are not available in this situation which considers the input graph as a long sequence composed of all nodes, we leverage the standard Transformer to learn the node representations.
>
> Due to the quadratic complexity on the number of nodes, this variant meets the Out-of-Memory problem on all datasets except the Photo dataset. And the performance is 94.18% on Photo, which is inferior to NAGphormer, whose performance is 95.49%. This is because compared to the straightforward construction, constructing a sequence of tokens enables the model to adaptively capture the importance of different-hop neighborhoods for each node.
>
> >Q3. "Mini-batch training for large-scale graphs has already been studied in the literature. Unless the authors present a new technique, which significantly improves scalability, It is hard to consider the improved scalability as the contribution of this paper."
>
> A3. Mini-batch training methods have been well-studied to improve the scalability of GNNs.
> However, these methods are not suitable for graph Transformers since existing graph Transformers are required to calculate the attention score for all node pairs and thus only support full-batch training, limiting their scalability for large-scale graphs. Hence, designing a novel graph Transformer that can handle large-scale graphs is essential and meaningful work for the community.
>
> This paper proposes NAGphormer that improves the scalability of graph Transformer by supporting the mini-batch training strategy via a novel module called Hop2Token, exhibiting promising performance on large-scale graphs.
> Therefore, we consider the improved scalability of graph Transformers as a main contribution of this paper.

---

### Author Response · Authors · 2022-11-15
**Summary of the revised version**

We sincerely thank all the reviewers for their positive and detailed assessments of our paper. We have revised our paper based on the reviews. Here, we list the main changes (highlighted in blue in the revised paper).

* Comparison of GraphGPS

    GraphGPS [1] is a recent work that reduces the complexity of the graph Transformer by utilizing the linear Transformer backbone. However, it still meets the Out-of-Memory problem on several dense or large graphs. We have added the experimental results of GraphGPS on six small-scale datasets in Table 1 of the revised version, which are inferior to NAGphormer.

* Experiments of the model’s complexity

    We have conducted experiments to evaluate the efficiency of NAGphormer in Appendix G. We compare the running time and GPU memory cost of NAGphormer and three scalable GNNs on large-scale graphs. The experimental results demonstrate the efficiency of NAGphormer.

* Updated results on large-scale datasets

    Due to the time limitation, we did not carefully search the parameters of NAGphormer on large-scale graphs in the original submission. In the revised version, following the suggestion of the reviewers, we do parameter study on a larger range for the propagation steps $k$. Then we do a more complete search for the parameters and have found the better parameters of NAGphormer that can achieve higher performance on large-scale graphs.

    Hence, we update the results of NAGphormer on large-scale datasets in the revised version, including Table 2, Table 3, Figure 3 and Figure 4.

* More related works

    We have added several recent works about GNNs and graph Transformers in the related works in the revised version.

* Proofreading

    We have proofread the paper and polished the overall writing.

[1] Ladislav Rampášek, et al. Recipe for a General, Powerful, Scalable Graph Transformer. NeurIPS 2022.

---

### Decision · Program_Chairs · 2023-01-20

**Decision:**

Accept: poster

**Justification For Why Not Higher Score:**

See the above-mentioned weaknesses.

**Justification For Why Not Lower Score:**

See the above-mentioned strengths.

**Metareview: Summary, Strengths And Weaknesses:**

This paper proposes a graph transformer architecture that aggregates the neighborhood features from different hops and treats them as a sequence of token vectors for the transformer. It shows improved expressive power with information from multi-hop neighborhoods. The authors theoretically analyze that the proposed method has more expressive power than decoupled GCN.

**Strengths:**
* The proposed method shows competitive performance on many benchmark datasets. The empirical performance of NAGphormer is outstanding.
* The multi-view/multi-scale representation of a node (a sequence of aggregated node features) is an interesting idea. The late fusion of the node features is new.

**Weaknesses:**
* One of the main contributions is "a sequence of tokens" for a node. Authors could provide additional ablation analyses. For instance, simple feature concatenation is one naïve approach. Compared to straightforward construction, the authors should show the performance gain of the sequence of token node representation.
* The authors argue that existing MPNNs have limitations such as over-smoothing and over-squashing. However, recent studies address those limitations, and it is unclear what the implications are for the NAGphormer. From the ablation study over the depth of the model, it seems that NAGphormer also suffers from over-smoothing or over-squashing problems since, as the depth increases, the performance deteriorates.

**Note From Pc:**

if the above contains the word "oral" or "spotlight" please see: "oral" presentation means -> notable-top-5% and "spotlight" means -> notable-top-25%. As stated in our emails, we are disassociating presentation type from AC recommendations